# A Multidisciplinary Approach in Examining the Susceptibility to Microbial Attack of Polyacrylic and Polyurethane Resins Used in Art Restoration

**DOI:** 10.3390/ijms231911725

**Published:** 2022-10-03

**Authors:** Raffaella Campana, Luigia Sabatini, Luca Giorgi, Giulia Pettinari, Laura Valentini, Pietro Gobbi

**Affiliations:** 1Department of Biomolecular Sciences, Division of Pharmacology and Hygiene, University of Urbino Carlo Bo, 61029 Urbino, Italy; 2Department of Pure and Applied Sciences, University of Urbino Carlo Bo, 61029 Urbino, Italy

**Keywords:** restoration, synthetic resins, microbial attack, ESEM, FTIR

## Abstract

The synthetic polymers used to protect artworks from deterioration process can be colonized by the fungi and bacteria responsible for the biodeterioration process. In this study, the susceptibility of synthetic polyacrylics and polyurethane resins to microorganisms (*Aspergillus niger* ATCC 9642, *Aureobasidium pullulans* ATCC 15233, *Chaetomium globosum* ATCC 6205, *Cladosporium cladosporioides* ATCC 16022, *Alternaria alternata* BC01, *Penicillium citrinum* LS1 and *Pseudomonas aeruginosa* ATCC 9027) was investigated. The microbial attack was simulated alone and with a biocide and the related growth was observed up to 21 days for bacteria and 28 days for fungi. The polyacrylic and polyurethane resins were subjected to microbial attack, regardless of the biocide treatment, with a fungal growth from 60% to the complete coverage of the plate surface. *Penicillium citrinum* showed the greatest adaptation ability and was found in all the examined resins. *P. aeruginosa* was visible in all the different resins, regardless of the presence of biocide. An environmental scanning electron microscope (ESEM) revealed the presence of fungal conidia and hyphae in the inoculated resins and the Fourier transform IR spectroscopy (FTIR-ATR) indicated chemical transformations in the IR spectra, particularly the hydrolysis of esters, with some differences between the polyacrylic and polyurethane resins, which were probably due to their different chemical features. Overall, our data stress that the chemical, physical and biological deterioration caused by microorganisms capable of degrading synthetic polymers is still a problem in art restoration and that new strategies must be considered to counteract this phenomenon.

## 1. Introduction

Cultural heritage and buildings are subjected to a natural and irreversible degradation process, which is determined by different factors, such as the chemical composition and nature of the material itself, the climate and exposure of the object, the frequency with which the surface is cleaned and the type and place of conservation [1]. The protection of t different artworks is a common and well consolidated practice performed by surface treatment with polymers that are able to form a protective layer on the object’s surface, as well as to control the transport of fluids, such as water, from the surface inside the object [2,3]. The restoration of art includes cleaning, a transient treatment to remove the materials not originally belonging to the artwork, and consolidation, a durable intervention that should remediate, prevent, or slow down further degradation due to ageing or external agents. The consolidation and protection of paintings, stone, wood, paper, glass, ceramic, and bones is possible by the simple application of organic solutions (through immersion, injection or brushing) of acrylic, vinylic, and silicone-based polymers and/or their copolymers. Most of these polymers are also considered reversible because, if necessary, the solvent used for the primary application can be used later for the removal of the resins [4]. However, although the application of these synthetic polymers slows down deterioration processes, it does not stop them. Therefore, it is important to consider the chemical and physical properties of polymers in order to evaluate their suitability for use in restoration treatments [5]. Major issues were recognized in the chemical stability and solubility of the commonly used solvents. Indeed, polymers’ deterioration in outdoor conditions can modify their chemical composition, leading to the formation of oxidized species, in turn causing the yellowing of the treated surfaces. In addition, the physical properties of polymers can change, inducing stiffening and brittleness, which often result in polymer fissures, detachments from the substrate and the worsening of the mechanical properties [6]. In recent years, it became evident that polymers have some disadvantages in the preservation process, such as the poor compatibility between organic conservation materials and inorganic heritage objects, as well as the degradation process [1,7,8].

Microorganisms, including bacteria and fungi, are among the factors responsible for the biodeterioration of cultural-heritage objects [9]. In particular, fungi can grow at relatively low levels of temperature and damp [10,11] and may be found in support materials (cellulose, canvas, wood, parchment, etc.), as well as in the materials used to fix paintings to their supports (animal or plant glues, oils, waxes and polysaccharides). In this sense, polymers used in artistic restoration can also be colonized by microorganisms and used as substrates for their growth [12]. Indeed, these materials are exposed to a variety of natural and artificial conditions (temperature, available water, cleaning chemicals and frequency, carbon sources and pollutants) that can result in their progressive deterioration and loss of functionality. In addition, microorganisms are able to produce and secrete two types of enzymes, intracellular and extracellular, which are involved in the transformation of organic compounds. In the available literature, the recent investigation by Zander et al. [13] reported important information on the susceptibility to microorganisms of polymers for different types of applications, adding data to the research previously presented by Capittelli and Sorlini [14], which specifically focused on the polymers used by art conservators.

From this perspective, the proposed work aimed to evaluate, through a multidisciplinary approach, the susceptibility to fungal and bacterial colonization of seven synthetic resins (four polyacrylics and three polyurethane) specifically used in artistic restoration. The microbial attack with different fungal species and bacteria was simulated on different resins, alone and with the addition of a biocide, and the related growth was monitored weekly up to 28 days for the fungi and 21 days for the bacteria. An environmental scanning electron microscope (ESEM) and Fourier transform IR spectroscopy in attenuated total reflection (FTIR-ATM) were applied to ascertain the microbial morphology and eventual alteration of the chemical composition of the resins, respectively.

## 2. Results

### 2.1. Resins’ Susceptibility to Microbial Attack

The polyacrylic polymers (AC 012, AC 013, AC P1 and AC 32) were subjected to microbial attack, as were the polyurethane polymers (PU H6, PU MW2 and PU GS) regardless of the biocide treatment (Figure 1). In the case of AC 012 and AC 013, the resins inoculated with fungi, as well as those treated with the biocide appeared almost completely covered by a green mycelium, a typical feature of *Penicillium* spp., after only 7 days of incubation (Figure 1a,b). A similar trend was noticed in the PU H6 and PU MW2 resins (Figure 1c,d), while cream-pink colonies related to the growth of *Aureobasidium pullulans* ATCC 15233 were evident in the PU GS sample (Figure 1e). The two last resins, AC P1 and AC 32, showed the diffuse fungal growth typical of *Penicillium citrinum* LS1 (Figure 1f,g). No colonies were observed in any of the non-inoculated control samples. For all the examined resins, the R was estimated as 4, indicating fungal growth from 60% to complete coverage of the surface. Indeed, the addition of biocide did not increase the resistance to microbial attack but, in some cases, partially inhibited the growth of *Penicillium*, thus favoring that of other species.

The *Pseudomonas aeruginosa* ATCC 9027 was visible in all the sub-cultures obtained from the different resins after 7 to 21 days of incubation, regardless of the presence of biocide (Table 1) (Figure 2).

### 2.2. ESEM

The ESEM observations allowed us to verify the presence of the different species on the surfaces of the examined resins, as well as their possible chemical variations. Representative ESEM images are shown in Figure 3. As a general trend, after 7 days of incubation, the growth of the inoculated fungi was observed in all seven resins. Specifically, the acrylic polymers (AC 012, AC 013, AC P1 and AC 32) presented conidia and conidiophores (Figure 3a,b) of *P. citrinum* LS1 (Figure 3d,e), while the polyurethane PU H6 also showed conidia of *A. pullulans* ATCC 15233 (Figure 3c). In the other two polyurethane resins (PU GS and PU MW2) conidia of *A. pullulans* ATCC 15233 (Figure 3f) and conidia and conidiophores of *P. citrinum* LS1 were clearly distinguishable (Figure 3g).

In the case of *P. aeruginosa* ATCC 9027, the presence of bacilli aggregates in all the examined resins was observed; these were more diffuse in AC 012 and AC 013 (Figure 4).

To examine the chemical variations of the resins, semi-quantitative analyses were performed. Many of the resins inoculated with fungi, but also several controls, revealed the presence of some elements in low percentages (Mg, P, S, K) (data not shown), in addition to the expected C, O and N. This was probably referred to the used culture medium (NSA).

### 2.3. FTIR Analysis

#### 2.3.1. Polyacrylic Resins (AC 012, AC 013, AC P1 and AC 32)

For the polyacrylic resins (AC 012, AC 013, AC P1 and AC 32), the IR spectra had the characteristic signal at 1727 cm^−1^, which was assigned to the stretching carbonyl ester vibration, and signals between 2980 and 2870 cm^−1^ due to the vibrations of the CH bonds present in the main chain of the macromolecule and in the alkyl residues bound to the ester oxygen atom. The area of the fingerprint had signals at 1450, 1390, 1240, 1150, 990, 840 and 750 cm^−1^ due to the stretching and bending vibrations of the bonds present in the molecule [15]. In particular, the 1150 cm^−1^ signal was attributed to the stretching of the ester C–O bond and shifted according to the type of alkyl remainder it carried. This signal was found at 1144 cm^−1^ for the resins AC 012 and AC P1, 1160 cm^−1^ for AC 013 and 1146 cm^−1^ for AC 32 (Figure 5 and Appendix A).

After incubation with microorganisms, the main chemical transformations in the IR spectra were constituted by the hydrolysis of esters, evidenced by the appearance of a carbonyl signal around 1650 cm^−1^ and a hydroxyl signal at 3300 cm^−1^, which was attributed to the presence of carboxylic groups (COOH) and carboxylate anions (COO–), eventually stabilized by hydrogen bond with vicinal carboxyl groups present in the chain (Appendix A). The hydrolysis of the esters produced a COOH carboxyl group, which remained connected to the polymer chain, and an alcohol group, which, depending on the type of polymer, could be methanol, ethanol, isopropanol, propanol, or butanol, a group that was not visible in the spectrum, probably because they were partly metabolized by the microorganism and partly volatilized.

Therefore, it was possible to measure the progress of the microbial attack by monitoring the growth over time of the intensity of the signals at 1650 and 3300 cm^−1^, indicating the formation of carboxylic groups caused by the hydrolysis of esters. In this sense, the fungi were able to determine a greater level of degradation than that caused by *P. aeruginosa* ATCC 9027 (Appendix A), as can be observed from the more marked variations of the spectra also involving the area of the fingerprint between 700 and 1500 cm^−1^ (Appendix A). Among the acrylics, the most resistant polymer was AC 012, for which the degradation products were detected in negligible quantities, while the least resistant was AC 32, whose structure was strongly altered by the presence of the microorganisms (Figure 5). In this case, the signals of the carboxyl groups were particularly intense, and heavy variations were also noted in the area of the fingerprint (between 700 and 1500 cm^−1^). By contrast, depolymerization phenomena were never observed, suggesting that the main chain of polyacrylic polymers was not attacked by microorganisms.

#### 2.3.2. Polyurethane Resins (PU H6, PU GS and PU MW2)

The IR spectra of the polyurethane polymers (PU H6, PU GS, PU MW2) had a characteristic signal around 1700 cm^−1^, assigned to the carbonyl stretching vibration of the carbamate group, located at 1714 cm^−1^ for PU H6 and 1700 cm^−1^ for PU GS, while this signal was found to be superimposed on that of the esters at 1730 cm^−1^ for PU MW2 [15]. The signals around 3300–3350 cm^−1^ and 1530 cm^−1^, assigned to the stretching and bending, respectively, of the secondary amidic N-H bond present in the carbamate group, were also clearly visible. In addition, the intense signal at 1100 cm^−1^ present in PU H6 and PU GS was attributable to the stretching of the ether C–O bonds present in the polyether chains that were distant from the carbamate groups. In the sample PU MW2, the signals at 1230 cm^−1^ were visible, and attributed to the stretching of the C–O ester bond. The signals between 2970 and 2860 cm^−1^ were due to the vibrations of the C–H bonds present in the main chain of the macromolecule (Appendix A).

After their incubation with the microorganisms, the polymers PU H6 and PU GS were not particularly altered, showing that the aliphatic and polyethylene polyurethane polymers were not clearly attacked. Indeed, the spectra, including the area of fingerprints, remained rather unchanged (Appendix A). In the case of PU H6, an aliphatic polymer, there was a slight variation in the pattern of the C–H bond signals and the disappearance of the signals at 1344 cm^−1^ and 930 cm^−1^. This was interpretable as an oxidative degradation of the aliphatic chain, but of too small entity to exert an influence on the physical properties (Figure 5). By contrast, the polyester-based polyurethane polymer, PU MW2, resisted the bacterial attack but underwent heavy alterations due to the fungal hydrolysis of the ester groups (Figure 5), which was evident from the appearance of the typical signals of carboxylic acids (3300 cm^−1^) and the relative decrease in the signal intensity of the esters at 1730 and 1230 cm^−1^.

## 3. Discussion

Natural polymers are generally susceptible to bacterial and fungal attack, while synthetic polymers can show different levels of susceptibility to microbial colonization due to their chemical nature [12,16,17]. Indeed, microorganisms can modify the structure and functionality of synthetic polymers in different ways through the biological coating of the surface, the production of metabolites (acids), enzymatic attack, physical penetration and subsequent disruption, possible water accumulation, and the production of pigments [18]. It is well established that molecular weight (MW) plays an important role in the microbial deterioration process; indeed, while polymers with high MW are less subject to microbial degradation, monomers, dimers and oligo-polymers are more susceptible to this type of attack [12]. In this regard, although the biological degradation of materials is difficult to reproduce in artificial conditions [19], laboratory tests are fundamental to assess the suitability of the products applicable to the restoration and to define their effectiveness and security for artworks [5]. Microbiological tests vary according to the type of product tested and, in the case of resins, such as adhesives, consolidating agents, protective products, etc., they have the purpose of verifying their degree of susceptibility to biological attack, as well as any alterations induced by microbial colonization.

In the present research, the susceptibility of synthetic polyacrylic and polyurethane resins, designed for artworks conservation and restoration, to attack by some microbial species was evaluated alone and also in the presence of a normally applied biocide (OIT, 15%), using a mineral salt medium (NSA), a medium specifically designed without any source of carbon to limit microbial growth around the applied resins. With regards to the resins not treated with OIT, consistent fungal development was found in all the samples, starting from the seventh day after the inoculum. It can be inferred that when a biomass starts to grow on an object (in our case, on the resin), the catabolic products of microorganisms grown first on a material could be used by other microorganisms not that are not directly capable of degrading the polymer. This condition, practically a second colonization, may lead to further remarkable damage by the microorganisms living on the surface of the material. In our case, *P. citrinum* was the fungal strain found in all the resins, regardless of the typology, except for PU GS, in which the presence of *A. pullulans* was also detected. It is conceivable that *Penicillium* possesses the greatest ability to adapt to the conditions provided by the examined resins. Indeed, this microorganism is frequently found on artworks, as well as on polyurethane and acrylic synthetic resins [20]. These data are in agreement with those available in the literature on the remarkable adaptability of fungi to the various environmental conditions that can be found in the field of applications of the synthetic polymers [21]. In particular, Cappitelli et al. [17] reported that marble samples treated with acrylic resins in the cathedral of Milano were subjected to biodeterioration by dematiaceous meristematic fungi, thus raising concern over the real effectiveness of the acrylics.

In the subsequent ESEM observation, all the examined resins became spotty. This could have been due to the presence of pigments on the walls of the fungal elements (such as the spores), or to the excretion of exo-pigments [22]. These molecules can be water-soluble and, therefore, once they leak from cells, they may spread in the medium. The alteration caused by pigments may be more or less relevant, depending on the bond they establish with the substrate, but the presence of colored spots is in any case undesirable. In addition, besides the presence of numerous spores, in some resins (such as PU H6 and AC P1) the formation of reproductive structures, indicative of fungal growth, was observed.

The development of an efficient protocol to counteract microbial attacks on cultural objects may follow two paths. The first is the removal of already existing biomass, usually through the application of biocides that are effective against a broad spectrum of microorganisms. The second is to prevent recolonization through the application of coating products in combination with repellents, surfactants or biocides [23]. In this research, we chose the second path, treating several of the examined resins with a homemade solution of 2-n-octlyl-4-isothiazolin-3-one (OIT 15%). Isothiazolinones (including 2-n-octlyl-4-isothiazolin-3-one) are used in occupational and industrial applications for their bacteriostatic and fungistatic activity. Despite their effectiveness as biocides, isothiazolinones are strong sensitizers and are responsible for skin irritations and allergies, as well as posing eco-toxicological hazards [24]. Our results indicated that this compound was not able to significantly inhibit the microbial growth, but limited the development of *P. citrinum* compared to that of *A. pulllans* and *Alternaria alternata*. It can be supposed that the biocide added to the resin partially inhibited the growth of *P. citrinum*, thus allowing the development of the other two less invasive species. In the literature, it is reported that the addition of biocides does not always prevent or delay the polymer-deterioration process. Shirakawa et al. [25] studied the fungal communities on acrylic paints treated with different biocides (carbamate, *N*-octyl-2H-isothiazolin-3-one, and *N*-(3,4-dichlorophenyl) -*N*-dimethylurea) and observed that the biocides did not affect the fungal biodiversity but, rather, mainly the abundance, without completely inhibiting the growth. Similarly, Abdel-Kareem [26] reported that, among fungicides tested on linen fabrics impregnated with synthetic resins, only dichlorophene was able to prevent microbial development, but showed the disadvantage of accelerating the deterioration provoked by light and heat. The study by Martin-Sanchez et al. [27] showed that the effectiveness of biocide treatments (such as quaternary ammonium, benzalkonium chloride, 2-octyl-2H-isothiazol-3-one, and Parmetol) progressively decreased over time, followed by new fungal growth. Another important consideration is that organic biocides can act as nutrients for microorganisms and, in this case, the choice of the biocide should be made on the basis of the results of experiments in the laboratory as well as in situ, because many factors can affect effectiveness, such as the dose, the interaction with the substrate and the sensitivities of different microbial species [28,29]. For these reasons, environmental regulations in Europe and elsewhere have restricted the use of biocides, which are considered potentially dangerous both for humans and the environment [14].

Microorganisms are able to use extracellular and intracellular enzymes to hydrolyze organic matter, transforming complex compounds into simple molecules that are easily assimilated [30]. In particular, fungi can break macromolecules using a set of hydrolytic enzymes, as well as monooxygenase, esterase, ureases and oxidoreductase [31,32]. Some of these exo-enzymes, produced constitutively at low levels, are responsible for the first breakdown of complex polymers into short chains or monomers [33]. Fungi are highly sensitive to varying concentrations of the substrate and can rapidly and dynamically respond to the changing availability of specific resources. In this regard, the concept of co-metabolism, defined as “the obligatory presence of a growth substrate or another utilizable compound critically needed to maintain biomass and induce the corresponding enzymes and/or cofactors for the biodegradation” [34,35], can be introduced as the potential metabolic system in the biodegradation process observed in the present research. A technique such as infrared spectroscopy offers the opportunity to monitor the chemical changes of different substrates and, in the literature, it is well reported that the biodeteriorated synthetic resins used in conservation can be distinguished from the same non-biodeteriorated polymers by the presence of a peptide bond in the infrared spectrum [16,17,18,36]. Indeed, in our research, the FTIR analysis indicated that the presence of the selected microorganisms (fungi and bacteria) induced chemical transformations in the IR spectra, with some differences between the polyacrylic and polyurethane resins. Interestingly, among the acrylics, AC 01 showed negligible degradation (hydrolysis of esters), while the structure of AC 32 changed significantly in the presence of the microorganisms. Similarly, considering the polyurethane resins, some were not particularly altered (such as PU H6 and PU GS), while others evidenced heavy alterations imputable to the fungal hydrolysis of the ester groups. These findings were in agreement with the observations reported by Cappitelli and Sorlini [14] suggesting that that biodeterioration of polyurethane polymers occurs through the enzymatic actions of hydrolases, such as ureases, proteases, and esterases. In addition, the investigation by McNamara et al. [37] found that a yeast isolated from a bronze statue covered with acrylic compound accelerated the deterioration of the coating. The degradation rate can be significantly affected by the chemical structure, particularly the C–C and other types of bonds, MW, structures and configuration, as well as the microorganisms involved [12]. In addition, the presence of other substances in the formulation, such as additives, may have influenced the interactions microorganism-substrate. In our case, we cannot exclude the possibility that the interaction between the examined fungi also led to the observed chemical changes in the different resins. In the literature, it was suggested that acrylics with long side chains were more susceptible to biodeterioration than homologous polymers with short side chains, probably as a consequence of the minor enzymatic steric impediment of compounds with long lateral chains [16,18]. Unfortunately, the impossibility of determining the real composition of the analyzed resins as well as of performing further chemical analysis has limited the complete understanding of the biodegradation process of the individual polymers. Based on our results, we can state that in art restoration and consolidation, alternative options, such as nanotechnologies and nanomaterials, particularly metal nanoparticles (gold, copper and silver), metal oxides (zinc, titanium, iron and aluminum) and tubular nanomaterials [38,39] can be considered for their physical, chemical and mechanical properties, as well as for their resistance to microbial attack. Indeed, once applied to the surfaces of damaged objects, these nanomaterials create a self-cleaning system, which is able to preserve the initial appearance of the treated elements, with a simultaneous decrease in the deposition of pollutants and a reduction in the external degradation due to contamination and microorganisms [39]. In the recent literature, polyacrylic acid (PAA), a non-toxic, biocompatible and biodegradable polymer, was reported as a candidate for so-called PAA nano-derivatives [40], with broad applications, including materials and manufacturing processes.

## 4. Materials and Methods

### 4.1. Polyacrylic and Polyurethane Resins

Seven synthetic resins (here indicated as AC 012, AC 013, AC P1, AC 32 for the polyacrylics and PU H6, PU GS, PU MW2 for the polyuretnanes) used in restoration practices were considered in this study. The manufacturer (ICAP Leather Chem, Lainate, Milan, Italy) preferred not to reveal information on the synthesis process, the composition or molecular weights of the provided products (defined as “confidential”). No further chemical or chemical-physical analyses were authorized. All the polymers were used in aqueous dispersion. A 15% aqueous solution of 2-n-octlyl-4-isothiazolin-3-one (OIT) (*v*/*v*) was prepared in our laboratory and used as biocide [41].

### 4.2. Microbial Strains and Culture Conditions

To simulate the microbial attack of the resins, six fungal strains were used in this study: *Aspergillus niger* ATCC 9642, *Aureobasidium pullulans* ATCC 15233, *Chaetomium globosum* ATCC 6205, *Cladosporium cladosporioides* ATCC 16022. All these fungi were purchased from the American Type Culture Collection (Manassa, VA, USA). *Alternaria alternata* BC01 and *Penicillium citrinum* LS1, previously isolated from wooden artwork [42], were included. The reference strain, *Pseudomonas aeruginosa* ATCC 9027, was also added. Fungi were grown on potato dextrose agar (PDA) (Liofilchem, Roseto degli Abruzzi, Italy) at 28 °C for 7 days, while *P. aeruginosa* ATCC 9027 was grown on tryptic soy agar (TSA) (VWR, Milan, Italy) at 37 °C for 24 h.

### 4.3. Inoculum Preparation

The different spore suspensions were prepared following the indication of the National Committee for Clinical Laboratory Standards [43]. Briefly, for each strain, spores were harvested from PDA plate by adding 2 mL of sterile 0.85% saline solution; the surface was gently scraped with a sterile spatula. The suspension was transferred in a sterile tube and left at room temperature for 5 min to allow the sedimentation of hyphal fragments. The upper homogeneous suspension was vortexed for 15 s and adjusted to an optical density at 530 nm, corresponding to about 10^6^ spores/mL. The quantification of each inoculum was verified (5 × 10^6^ spores/mL) with the agar-plate-count method on PDA. To perform the microbial attack, equal volumes of each spore suspension were blended to obtain the final mixed spore suspension.

In the case of *P. aeruginosa* ATCC 9027, several colonies were inoculated in 10 mL of sterile Mueller–Hinton broth (MHB) (Oxoid) and incubated at 37 °C for 24 h. At the end of incubation, the bacterial suspension was sprectrophotometrically adjusted to an optical density (610 nm) of 0.15, corresponding to about 10^6^ cfu/mL.

All the inoculums were used for the following microbiological tests.

### 4.4. Simulated Microbial Attack

The simulated microbial attack was performed on nutrient salt agar (NSA), composed as follows: KH_2_PO_4_ 0.7 g/L, MgSO_4_·7H_2_O 0.7 g/L, NH_4_NO_3_ 1 g/L, NaCl 0.005 g/L, FeSO_4_·7H_2_O 0.002 g/L, ZnSO_4_·7H_2_O 0.002 g/L, MnSO_4_·H_2_O 0.001 g/L, K_2_HPO_4_ 0.7 g/L, agar 15 g/L (final pH 6.0–6.5). The NSA was then sterilized by autoclaving at 121 °C for 15 min. The seven above-mentioned resins were tested as films of about 50 × 25 mm (thickness about 0.2 mm), distributed on the surfaces of NSA plates, following the standard procedures [44,45] with some modifications. The resins were left to dry for 96 h at room temperature under aseptic conditions and were tested with and without the addition of the selected biocide (OIT 15%) on the surfaces of the different samples in a volume of about 100 µL. To simulate the microbial attack, the test specimens were inoculated with 75 μL of mixed spore suspension (obtained as described above) and incubated at 28 °C for 28 days. Non-inoculated resins and resins treated with biocide and inoculated with fungi were added as controls. All the specimens were regularly observed by the naked eye to check the fungal growth progression following the determined rating (R) [43]: R = 0, no visible growth; R = 1, trace of growth (less than 10%); R = 2, light growth (ranging from 10 to 30%); R = 3, medium growth (ranging from 30 to 60%); R = 4, heavy growth (ranging from 60% to complete coverage). All the notes about the visual fungal growth were recorded and, when necessary, a microscopic observation was performed (OLYMPUS CX 41).

In the case of *P. aeruginosa* ATCC 9027, 500 µL of bacterial suspension (about 10^5^ cells/mL) was inoculated directly into melted and cooled agar. Next, NSA was poured into petri dishes and allowed to solidify; at this point, the resins were distributed on the surface of NSA as described above and incubated at 37 °C for 21 days. Since the growth of *P. aeruginosa* ATCC 9027 on the resins was not visible to the naked eye, weekly, each resin was lightly touched with a sterile cotton swab subsequently streaked on cetrimide agar (VWR). The plates were incubated at 37 °C for 24 h to verify the occurrence of the typical *P. aeruginosa* pigmented blue-green colonies.

### 4.5. Environmental Electron Scanning Microscope (ESEM)

The samples were deposited into aluminum-specimen stubs, which were previously covered with a conductive carbon adhesive disk (TAAB Ltd., Calleva Park, Aldermaston, Berks, UK). A FEI Quanta 200 FEG Environmental Scanning Electron Microscope (FEI, Hillsboro, OR, USA), equipped with an energy-dispersive X-ray spectrometer (EDAX Inc., Mahwah, NJ, USA), was used to evaluate the microbial growth and physical modifications of the resins. The analyses were performed by using a focalized electron beam at a vacuum-electron-gun pressure of 5.0 × 10^−6^ mbar. The ESEM was utilized in low-vacuum mode, with a specimen chamber pressure set from 0.6 to 0.80 mbar, an accelerating voltage of 15–20 kV, and a magnification ranging between 700 and 5000×. The images were obtained by means of a back-scattered electron detector. The spectrometer unit was equipped with an ECON (Edax Carbon Oxigen Nitrogen) 6 utw X-ray detector and Genesis Analysis software. Each sample was analyzed with a time count of 100 s and an amp time of 51, while the probe current was 290 μA.

### 4.6. Fourier-Transform–Infrared-Spectroscopy (FT-IR)

The Fourier transform–infrared was applied to determine whether the microbial attack caused an alteration in the chemical composition of the considered resins. For this, FT-IR spectra of the seven resins inoculated with fungi or bacteria were acquired in attenuated total reflectance mode (ATR) using a Perkin–Elmer Spectrum Two FT-IRTM equipped with a ZnSe crystal. Resins not inoculated with microorganisms were used as controls. The sample was kept in contact with the crystal at controlled pressure and the spectrum was acquired with 4 scan. Air background was first acquired.

## 5. Conclusions

The applied multidisciplinary approach, including microbiological tests, morphological observations and a chemical analysis, evidenced that all the examined resins were subjected to microbial attack, regardless of typology or biocide treatment. Unfortunately, the lack of a clear indication of the constituent components of the used resins made it extremely difficult to predict the value and effectiveness of these materials over time and in different environments. In our experience, the question of the effectiveness of synthetic resins in protecting artworks from microbial attack is significant, and, for this reason, the assessment of their suitability in terms of antimicrobial activity is recommended. Cooperation among experts, such as microbiologists, materials scientists and conservators might be important for proposing novel approaches and more suitable tools for the restoration of items of our cultural heritage.

## Figures and Tables

**Figure 1 ijms-23-11725-f001:**
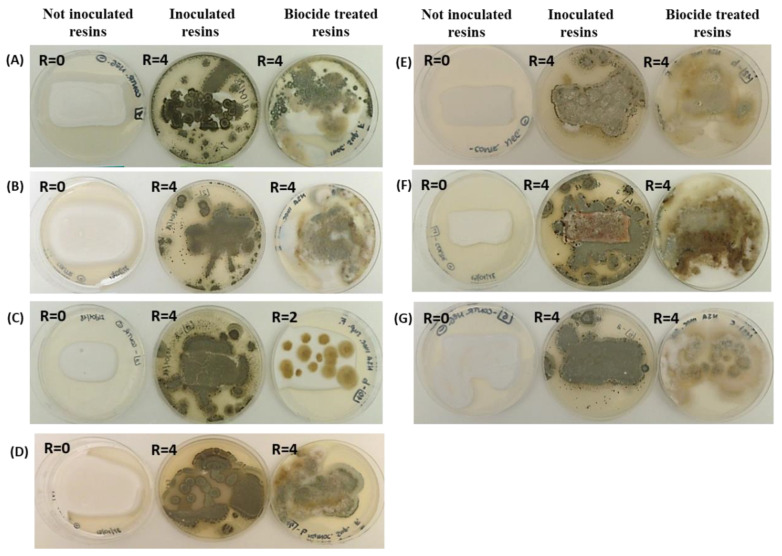
Representative visual observations of the fungal growth after 7 days of incubation on the surfaces of seven different polymers: (**A**) AC 012, (**B**) AC 013, (**C**) PU H6, (**D**) PU MW2, (**E**) PUGS, (**F**) ACP1, (**G**) AC 32. For each sample, non-inoculated control resins, resins inoculated with fungi, and resins treated with biocide and inoculated with fungi were considered. The R rating values (from 0 to 4) are also indicated.

**Figure 2 ijms-23-11725-f002:**
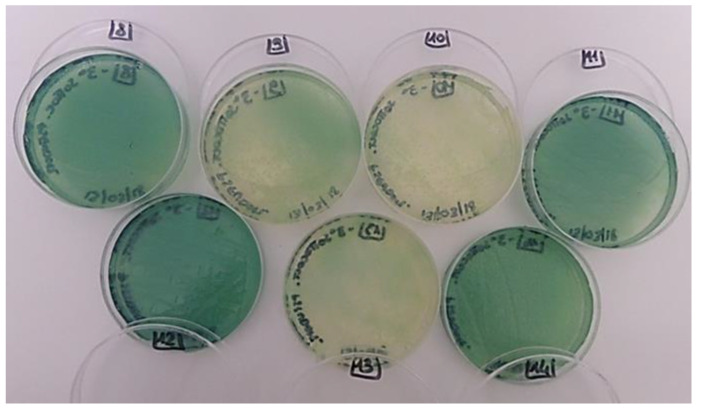
Representative visual observations of the positive cultures in cetrimide agar of *P. aeruginosa* ATCC 9027 from each inoculated resin.

**Figure 3 ijms-23-11725-f003:**
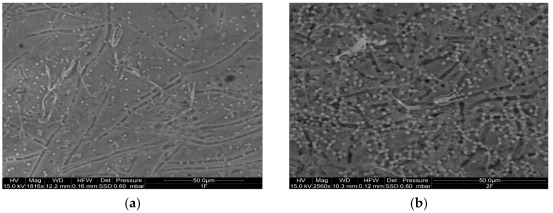
Representative ESEM images of the different resins with fungal growth recovered after 7 days of incubation: (**a**) AC 012 and (**b**) AC 013 presenting conidia and hyphae of *P. citrinum* LS1; (**c**) PU H6 with conidia, as well as conidiophores of *P. citrinum* LS1; (**d**) AC P1 with conidia and conidiophores of *P. citrinum* LS1; (**e**) AC 32 showing spores and hypae of *P. citrinum* LS1; (**f**) PU GS with contemporary presence of spores and hypae of *P. citrinum* LS1 and blastoconidia of *A. pullulans* ATCC 15233; (**g**) PU MW2 presenting conidia and conidiophores of *P. citrinum* LS1.

**Figure 4 ijms-23-11725-f004:**
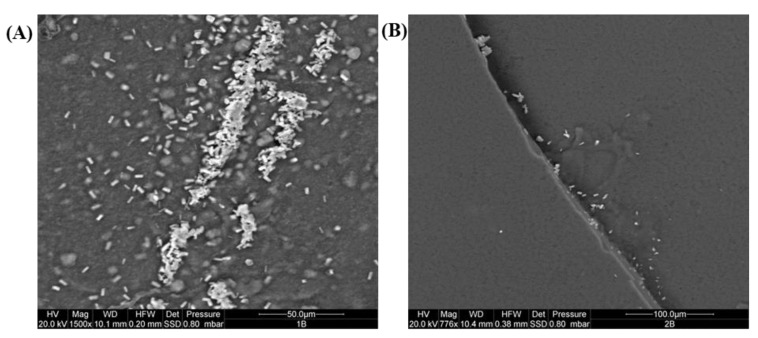
Representative ESEM images of AC 012 (**A**) and AC 013 (**B**) resins with evident bacilli growth on their surface after 7 days of incubation.

**Figure 5 ijms-23-11725-f005:**
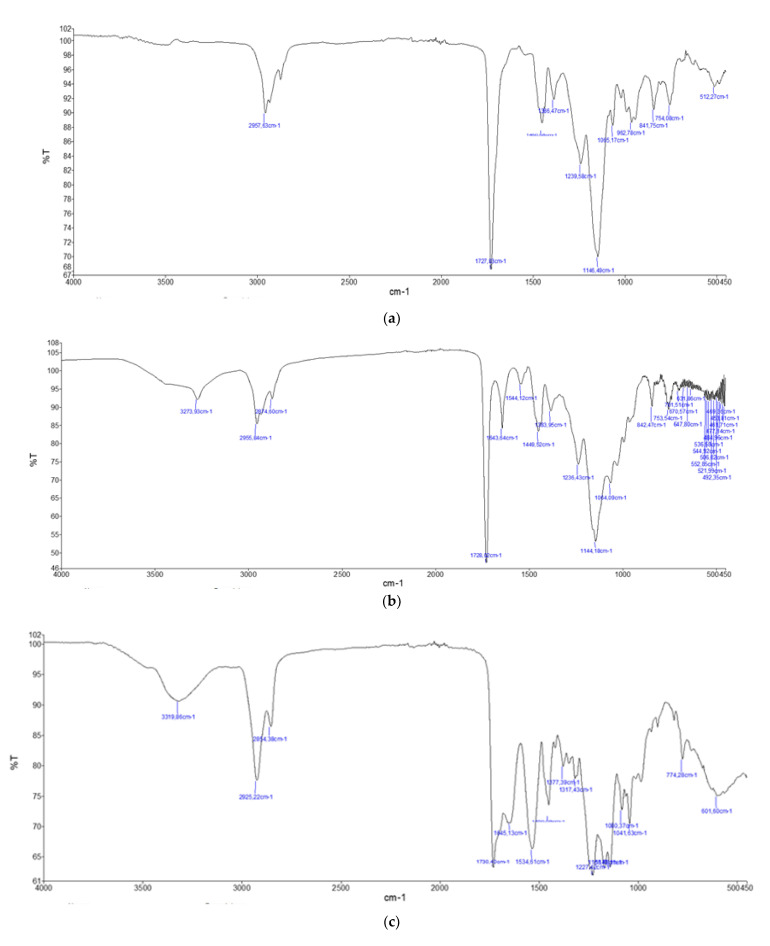
FT-IR spectra of polyacrylic resin AC 32 and polyurethane PU MW2 controls (**a**,**b**) and after 7 days of incubation with fungi (**c**,**d**). FT-IR spectra of polyurethane resin PU H6 control (**e**) and after 7 days of incubation with *P. aeruginosa* ATCC 9027 (**f**).

**Table 1 ijms-23-11725-t001:** Growth of *P. aeruginosa* ATCC 9027 in the subcultures on cetrimide agar from the different resins treated and not treated with the biocide (OIT, 15%) 7, 14 and 21 days after inoculation.

Resins	Growth of *P. aeruginosa* ATCC 9027		
	7 Days	14 Days	21 Days
AC 012 with biocide	Visible growth	Visible growth	Visible growth
AC 012 without biocide			
AC 013 with biocide	Visible growth	Visible growth	Visible growth
AC 013 without biocide			
PU H6 with biocide	Visible growth	Visible growth	Visible growth
PU H6 without biocide			
PU GS with biocide	Visible growth	Visible growth	Visible growth
PU GS without biocide			
PU MW2 with biocide	Visible growth	Visible growth	Visible growth
PU MW2 without biocide			
AC P1 with biocide	Visible growth	Visible growth	Visible growth
AC P1 without biocide			
AC 32 with biocide	Visible growth	Visible growth	Visible growth
AC 32 without biocide			

## Data Availability

The data presented in this study are available on request from the corresponding author.

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
