# Peer review of "A Multidisciplinary Approach in Examining the Susceptibility to Microbial Attack of Polyacrylic and Polyurethane Resins Used in Art Restoration"

_ijms, 2022, doi:10.3390/ijms231911725_

Round 1

Reviewer 1 Report

The authors present a manuscript on the colonisation of bacteria and fungi on proprietary polymeric materials used in art restoration. The manuscript was mostly well written and logical in its presentation. However, it would benefit from further proof-read to correct minor grammatical and typographical errors throughout the manuscript.

The main finding that bacteria and fungi grows on all polymers tested with and without the biocide, while not surprising, is interesting. Has the biocide (OIT) previously been evaluated for activity in solution against the tested strains of bacteria and fungus? The rationale to use a biocide that is known to be bacteriostatic and fungistatic should be further explained in the text. Why not have bactericidal and fungicidal biocides tested? It is also unclear from the methods how the biocide was applied, i.e. how long was the contact time, was it present at 15% concentration (w/v?) for the duration of the experiment?

Another limitation of the study was the use of proprietary resins, the composition of which are not disclosed. While the authors were upfront in disclosing this, this limits the impact of the study and its applicability to future comparisons.

Figures showing the IR spectrums are not illustrative of the changes discussed in the main text, texts/numbers in the figures are illegible. Consider removing the figure to supplementary section and enlarge the relevant range of the spectrum (for example, 1650 and 3300 cm-1) for the main text to illustrate the main findings in the results section.

Author Response

We appreciate the observations and we have tried to respond to the proposed questions in an adequate manner.

Has the biocide (OIT) previously been evaluated for activity in solution against the tested strains of bacteria and fungus? Why not have bactericidal and fungicidal biocides tested? It is also unclear from the methods how the biocide was applied, i.e. how long was the contact time, was it present at 15% concentration (w/v?) for the duration of the experiment?

Our responses: the used biocide (OIT) was directly suggested by the restorers of our University based on their experiences on different artistic objects. Data sheets of the product (from different manufacturers) report its efficacy in a wide range of applications including adhesives, polymer emulsions, decorative wood stains, sealants, pulp, and fountain solutions, also showing activity against fungi and bacteria in the wet state (Hopeland Chem-Tech Co; Iro Chemical). We are aware that, in our research, no data were presented on its activity against the selected fungi and bacteria, relating it to references on this topic.

As regards the methodology in our investigation, the solution was applied to the seven resin films in a volume able to cover the whole surface (about 100 µl) before the addition of the microorganisms. This information was added in the Experimental section.

The observed inefficacy of this biocide could be related to the lack of wet state of the resins themselves during the experiment (up to 28 days), rather than to the used concentration. Moreover, as indicated in the Discussion, the application of a biocide is not always successful: “Shirakawa et al. [25] have studied the fungal communities on acrylic paints treated with different biocides (carbamate, N-octyl-2H-isothiazolin-3-one, and N- (3,4-dichlorophenyl) -N-dimethylurea) and have observed as the biocides have not affected the fungal biodiversity but mainly the abundance, without completely inhibiting the growth. Similarly, Abdel-Kareem [26] reported that, among fungicides tested on linen fabrics impregnated with synthetic resins, only dichlorophene was able to prevent microbial development, but showing the disadvantage to accelerate the deterioration provoked by light and heat. The study of Martin-Sanchez et al. [27] showed that the effectiveness of biocide treatments (such as quaternary ammonium, benzalkonium chloride, 2-octyl-2H-isothiazol-3-one, and Parmetol) progressively decreased over time, followed by a new fungal growth.” 

Another limitation of the study was the use of proprietary resins, the composition of which are not disclosed. While the authors were upfront in disclosing this, this limits the impact of the study and its applicability to future comparisons.

Our response: we are completely agreeing with this observation and, unfortunately, the lack of clear indication of the constituent components of these products makes it extremely difficult to be able to predict the value of these materials over time and in different environments. This concept was added in the Conclusions.

Figures showing the IR spectrums are not illustrative of the changes discussed in the main text, texts/numbers in the figures are illegible. Consider removing the figure to supplementary section and enlarge the relevant range of the spectrum (for example, 1650 and 3300 cm-1) for the main text to illustrate the main findings in the results section.

Our response: we understand the question and we have decided to maintain in the main text only the most relevant images (the ones with visible chemical changes), moving all the others in Supplementary Materials.

Reviewer 2 Report

Review of the manuscript ID ijms-1928985 and entitled "A multidisciplinary approach for examining the susceptibility to microbial attack of polyacrylic and polyurethane resins used in art restoration".

In my opinion, the subject of the work is interesting and very important. I see the obtained results as a very important verification of the effectiveness of synthetic resins in protecting artworks against microbial attack.

I agree with the authors' opinion that "...the question of the effectiveness of synthetic resins in protecting artworks from the microbial attack raises, and for this reason, the assessment of their suitability in terms of antimicrobial activity is recommended. The cooperation among experts, such as microbiologists, materials scientists and conservators might be important for propose novel approaches and suitable tools applicable in items of our cultural heritage...”.

In my opinion, the manuscript is well written and its sections are consistent. I have no comments to the content of the manuscript. There is always room to do better (e.g. use more modern techniques). However, in this case, I do not see any such need. Therefore, I recommend the manuscript for publication in IJMS.

Author Response

We really thank the reviewer for the positive comments on the performed research.

Reviewer 3 Report

The paper entitled “A multidisciplinary approach for examining the susceptibility to microbial attack of polyacrylic and polyurethane resins used in art restoration” prepared by Campana et al evaluate by a multidisciplinary approach the susceptibility to fungal and bacterial colonization of seven synthetic resins (four polyacrylics and three polyurethane) specifically used in the artistic restoration. The work is interesting and the results support the claims. By the way, I recommend a major revision and it is needed to address the following issue:

And the last sentence in the conclusion should be excluded

In the experimental part, more details should be given

Line 351 - 353, please revise these sentences and write in proper manners

the aim and objective are not very clear. Please revise and make it clear

and concise.

please revise the sentences and remove the grammatical errors

what is the suggestion of this study for future works?

There are many studies investigating the importance of the topic , Please add these references to your introduction and discussion parts of the manuscript and compare and bold your study novelty: 10.1007/s10311-020-01073-y, https://doi.org/10.1016/j.soildyn.2022.107419, https://doi.org/10.3390/polym14061259

There are some spelling errors and logic problems in the text that need attention. Moreover, the typos in the manuscript need to be double-checked.

 It is better to compare the results of the present paper with previous works.

Results have not been properly/sufficiently interpreted in the discussion.

The conclusion needs to be upgraded may be Including a discussion part with the cost, and possible side effects or limitations in the use of these polymers to increase the impact of the paper.

Author Response

We appreciate the suggestions and we have tried, where possible, to respond correctly to these.

The last sentence in the conclusion should be excluded

Our response: in our opinion, the concept “The cooperation among experts, such as microbiologists, materials scientists and conservators might be important for proposing novel approaches and more suitable tools applicable in items of our cultural heritage” is well focused on the topic of the research and was appreciated by the others reviewers. Moreover, the lack of data constituting the examined resins suggests the need for transparency to be adopted in industrial settings, highlighting that this lack represents the central point of our manuscript. Thus, we have decided to maintain it.

In the experimental part, more details should be given

Our response: the experimental part is extremely detailed. In any case some further information are given.

Line 351 - 353, please revise these sentences and write in proper manners; the aim and objective are not very clear. Please revise and make it clear and concise.

Please revise the sentences and remove the grammatical errors

what is the suggestion of this study for future works?

Our response: we tried to follow the reviewer's suggestions and this part was modified and even corrected, adding a new reference.

There are many studies investigating the importance of the topic, Please add these references to your introduction and discussion parts of the manuscript and compare and bold your study novelty: 10.1007/s10311-020-01073-y,https://doi.org/10.1016/j.soildyn.2022.107419,  https://doi.org/10.3390/polym14061259

 Our response: see comment above.

There are some spelling errors and logic problems in the text that need attention. Moreover, the typos in the manuscript need to be double-checked.

 Our response: we have checked the grammar as well as the typos.

 It is better to compare the results of the present paper with previous works.

 Results have not been properly/sufficiently interpreted in the discussion.

Our response: we are not agreeing with the reviewer on these observations; in our opinion, the obtained results were properly discussed and compared with the adequate references on the topic.

The conclusion needs to be upgraded may be Including a discussion part with the cost, and possible side effects or limitations in the use of these polymers to increase the impact of the paper.

Our response: we appreciate this comment. We think that the examined polymers are not suitable for art consolidation as they resulted subjected to microbial contamination, thus alternatives should be considered (as indicated in the final part of the discussion). This already represents by self a great limitation for their use. The question of the cost and the possible side effects were not discussed because the research is mainly focused on the microbiological aspect.

Round 2

Reviewer 1 Report

The authors have made the necessary changes to address the reviewers comments.

Author Response

We thank the reviewer for the indications aimed to improve the quality of the presented research.

Reviewer 3 Report

Unfurtuantly authors didnt address all my comments as I asked before to discuss the following references to compare with similar papers. so I asked again please properly address this comment:

There are many studies investigating the importance of the topic , Please add these references to your introduction and discussion parts of the manuscript and compare and bold your study novelty: 10.1007/s10311-020-01073-y, https://doi.org/10.1016/j.soildyn.2022.107419, 

Author Response

Our response: We are not agreeing with the reviewer because two of the above mentioned references are not pertinent to our research.

Specifically, the article “Effective removal of the rare earth element dysprosium from wastewater with polyurethane sponge-supported graphene oxide–titanium phosphate. Environ Chem Lett 19, 719–728 (2021)” is focused on “nanomaterials as promising adsorbents to recover rare earth elements from wastewater” If the reviewer intended this article as a possible application of nanomaterials, the citation is right, but in the contest of art restoration is not well addressed.  

The second article “Effects of activated carbon on liquefaction resistance of calcareous sand treated with microbially induced calcium carbonate precipitation. Soil Dynamics and Earthquake Engineering 161, 107419, (2022)” is an interesting article focused on “Microbially induced calcium carbonate precipitation (MICP) is a new, natural, environmentally friendly treatment method for soil foundations, which can improve the resistance of calcareous sand foundations to liquefaction. Liquefaction is prone to occur under dynamic loading caused by events such as earthquakes or sea waves”, again non inherent the item of our research.

The third article was already added in the manuscript, Discussion, lines 360-363In the recent literature, polyacrylic acid (PAA), a non-toxic, biocompatible and biodegradable polymer, is reported as a candidate for the so-called PAA nanoderivatives [41], with broad applications, including materials and manufacturing processes

As regards the impact of our research, our data “stress that the chemical, physical and biological deterioration caused by microorganisms able of degrading synthetic polymers is still a problem in art restoration and new strategies must be considered to counteract this phenomenon”, confirming the question of the “the effectiveness of synthetic resins in protecting artworks from the microbial attack

Round 3

Reviewer 3 Report

the paper can be accepted